# Pruning nnU-Net with Minimal Performance Loss

**Tongyun Yang**                                        TONGUYUNYANG@TUDELFT.NL
**Yidong Zhao**                                         Y.ZHAO-8@TUDELFT.NL
**Qian Tao**                                            Q.TAO@TUDELFT.NL
*Department of Imaging Physics, Delft Unversity of Technology*

**Editors:** Accepted for publication at MIDL 2025

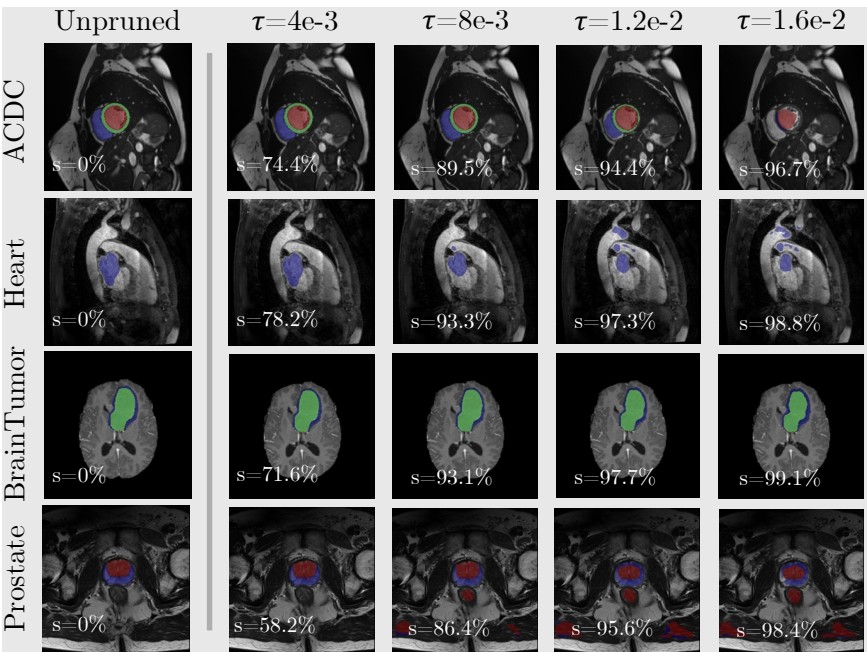

Figure 1: Comparison of the segmentation maps produced by the nnUNet pruned at various magnitude thresholds $\tau$, for four different datasets. Each column corresponds to a different value of $\tau$, and $s$ in each subfigure indicates the resulting weight sparsity.

## Abstract

nnU-Net is widely known for its accurate and robust segmentation performance in medical imaging tasks. However, the trained networks are typically heavily parameterized, and the high computational demand limit their deployment on devices with constrained resources. In this paper, we show that more than 80% of the trained nnU-Net weights can be removed without significant performance degradation, maintaining a proxy Dice score of $> 0.95$. This applies to both 2D and 3D configurations across four different medical image segmentation datasets. Interestingly, we observe that critical weights consistently concentrate near the U-Net encoder and decoder ends, while the bottleneck layers can be heavily pruned. These findings highlight the significant weight redundancy in trained nnU-Net and suggest opportunities for further optimization, to facilitate deployment of the model on devices with limited resources.

**Keywords:** Medical image segmentation, nnU-Net, weight pruning, sparsity

## 1. Introduction

nnU-Net (Isensee et al., 2021) has become the go-to architecture for medical image segmentation tasks, known for its exceptional out-of-the-box performance in a wide range of datasets. Despite its robustness, nnU-Net remains computationally intensive, which limits its feasibility for deployment in devices with restricted resources (Stock et al., 2024). Pruning is a widely used technique that introduces sparsity by removing less significant weights, successfully reducing computational demands in other domains such as computer vision (Tan and Le, 2019) and large language models (Frantar and Alistarh, 2023). Some recent works investigated pruning of the UNet architecture, for example, STAMP (Dinsdale et al., 2022) demonstrated that 85% sparse U-Net could outperform in data-scarce scenarios. Sauron U-Net (Valverde et al., 2024) employed channel pruning based on feature maps and achieved inference speed-ups. However, little effort has been directed towards studying the weight redundancy in the nnU-Net, which is often the default choice, and is auto-configured for any 2D and 3D medical image datasets.

To explore the potential of reduced memory and computational footprint for nnU-Net, in this empirical study, we investigate the weight redundancy of trained nnU-Nets for both 2D and 3D medical image datasets, and examine the cross-layer distribution of those potentially redundant weights in the original nnU-Net configuration.

## 2. Method

We develop an easily extendable pruning pipeline on top of the nnU-Netv2 pipeline. This pipeline fully leverages the standard nnU-Net preprocessing, postprocessing, and ensembling strategies. We employ post-training pruning, by first training the full model to convergence, and then selectively removing weights based on magnitude. Specifically, for each of the five cross-validation folds, we set weights with magnitude below a fixed threshold $\tau$ to zero: $w \leftarrow 0$ if $|w| < \tau$, a straightforward yet highly successful pruning strategy (Han et al., 2015). The pruned network is directly used without retraining.

To evaluate the efficacy of our pruning method, we report the segmentation performance using a *proxy* Dice score, defined as the Dice similarity coefficient (DSC) between predictions from the pruned and unpruned models. This proxy is used because many existing benchmarks do not provide ground truth for the testing set. By using the full nnU-Net as reference, we can effectively measure performance deviation after pruning.

## 3. Experiments

### 3.1. Implementation Details

All experiments were conducted on an NVIDIA RTX 4090 card with 64GB RAM. We evaluated on four datasets: Brain Tumor, Prostate, and Heart from the medical segmentation decathlon (MSD) challenge (Antonelli et al., 2022), and the ACDC dataset (Bernard et al., 2018). The nnU-Net auto-configuring pipeline resulted in models of different sizes: 155M parameters for Brain Tumor, 222M for Prostate, 103M for Heart, and 167M for ACDC, respectively. The code of the pruning pipeline [1] will be made publicly available.

---

1. https://github.com/prunennunet/Prune_nnUNet

### 3.2. Results

Figure. 2 (a) reports the segmentation performance of the model after pruning, showing that nnU-Net can be pruned by 80% with minimal performance degradation, i.e. maintaining 0.95 proxy DSC. For the ACDC dataset, which has the ground truth available for the testing set, we also plotted the true Dice score (in pink). Figure. 2 (b) shows the number of weights through the layers and their corresponding pruning ratios. It can be observed that while the parameter count increases in the deeper layers, these deeper layers exhibit a greater sparsity after pruning. This trend is consistent in all four datasets.

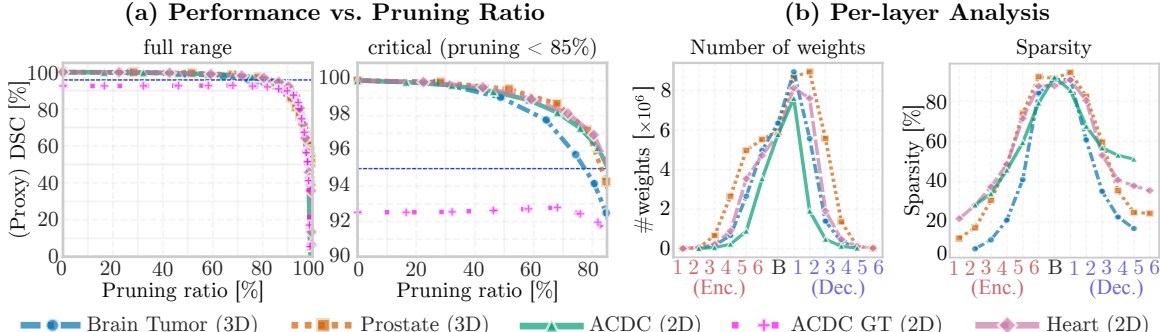

Figure 2: (a): Proxy DSC vs. pruning ratio across all datasets, showing minimal performance degradation up to 80% pruning. For the ACDC dataset where the ground truth (GT) segmentation is available for the testing set, the real DSC score is also plotted (in pink). (b): Weight size and pruning ratios across layers in the encoder (Enc.), bottleneck (B), and decoder (Dec.), revealing increasing sparsity in deeper layers and the bottleneck. The same trend is observed in all datasets.

## 4. Conclusion

Our results demonstrate that nnU-Net can retain competitive performance after aggressive pruning of up to 80% of weights using a straightforward magnitude-based method. This highlights a substantial weight redundancy in trained nnU-Nets. We further revealed an intriguing inverted-U-shaped weight redundancy pattern, consistent across four datasets of 2D and 3D medical images. Our findings suggest opportunities for further optimization of the deep learning network, to reduce the energy footprint and enable deployment of medical image segmentation on devices with limited resources.

## 5. Acknowledgment

We acknowledge the Dutch Research Council (NWO) and the National Growth Fund project AiNed for financial support.

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
