# OpenReview forum: "Pruning nnU-Net with Minimal Performance Loss"
_MIDL.io/2025/Short_Papers — MIDL 2025 - Short Papers_

### Official Review · Reviewer_k4fz · 2025-04-22

**Rating:** 3
**Confidence:** 3

**Summary:**

The paper presents a method to prune weights of a popular segmentation model nnU-Net, without substantially degrading the segmentation performance as compared to the model with full weights, or to provided masks. The (proxy) Dice on four datasets shows little degradation for even high levels of pruning.

**Strengths:**

-	Carbon footprint of methods is important due to larger and larger models being developed
-	Four different datasets are used
-	The methodology is simple to describe and grounded on existing popular work
-	Insightful analysis in Fig. 2

**Weaknesses:**

-	Given the aim of the paper, it would be relevant to at least give some ballpark estimation of the time/cycles/etc it takes for a nn-UNet to train vs evaluate, and how this is reduced with more sparse networks
-	There are many segmentation datasets available, it could have made sense to either select others with ground truth, or to simply subsample part of the training set \
-	It is difficult to appreciate Fig. 1 without knowing (proxy) Dice, for brain tumor the results look visually similar, but for the others there appears to be either over- or undersegmentation. Also it is unclear how the tau’s were picked here (they seem to be rather fine-grained)
-	I am not able to see the submission form anymore, but I believe MIDL encourages the code to be made available straight away so it can be reviewed

---

### Decision · Program_Chairs · 2025-05-01

Accept